# Splenic Volume as a Surrogate Marker of Immune Checkpoint Inhibitor Efficacy in Metastatic Non Small Cell Lung Cancer

**DOI:** 10.3390/cancers13123020

**Published:** 2021-06-16

**Authors:** Loïck Galland, Julie Lecuelle, Laure Favier, Cléa Fraisse, Aurélie Lagrange, Courèche Kaderbhai, Caroline Truntzer, François Ghiringhelli

**Affiliations:** 1Platform of Transfer in Biological Oncology, Georges François Leclerc Cancer Center—UNICANCER, 21000 Dijon, France; lgalland@cgfl.fr (L.G.); jlecuelle@cgfl.fr (J.L.); ctruntzer@cgfl.fr (C.T.); 2University of Burgundy-Franche Comté, Maison de l’université Esplanade Erasme, 21000 Dijon, France; 3Department of Medical Oncology, Georges François Leclerc Cancer Center—UNICANCER, 21000 Dijon, France; lfavier@cgfl.fr (L.F.); cfraisse@cgfl.fr (C.F.); alagrange@cgfl.fr (A.L.); CGKaderbhai@cgfl.fr (C.K.); 4Genomic and Immunotherapy Medical Institute, Dijon University Hospital, 21000 Dijon, France; 5UMR INSERM 1231, 21000 Dijon, France

**Keywords:** lung cancer, immunotherapy, splenomegaly, prognostic biomarkers

## Abstract

**Simple Summary:**

Monoclonal antibodies targeting PD1/PD-L1 are game changers in the treatment of advanced non-small cell lung cancer (NSCLC), but biomarkers are lacking. We previously reported the prognostic role of splenic volume in digestive cancer and its correlation with the presence of immunosuppressive cells. The aim of this study was to evaluate the prognostic role of splenic volume in NSCLC patients treated with immune checkpoint inhibitors (ICIs).

**Abstract:**

Monoclonal antibodies targeting PD1/PD-L1 are game changers in advanced non-small cell lung cancer (NSCLC), but biomarkers are lacking. We previously reported the prognostic role of splenic volume in digestive cancer and its correlation with the presence of immunosuppressive cells. The aim of this study was to evaluate the prognostic role of splenic volume in NSCLC patients treated with immune checkpoint inhibitors (ICIs). We conducted a retrospective study of 276 patients receiving ICIs for advanced NSCLC in the Georges François Leclerc Cancer Center. The association between splenic volume at baseline and at two months of therapy and progression-free survival (PFS) during ICI treatment or overall survival (OS) from ICI initiation was evaluated using univariate and multivariable Cox analyses. Splenic volume during treatment and the change in splenic volume were associated with poor PFS (respectively *p* = 0.02 and *p* = 0.001) and with OS (respectively *p* < 1.10^−3^ and *p* < 1.10^−3^). Baseline splenic volume at the first evaluation was also associated with poor OS (*p* = 0.001). LDH rate and dNLR were positively correlated with splenic volume, as well as with its evolution. After the adjustment of clinical variables, splenic volumes remained a predictive marker of immunotherapy efficacy. Splenic volume is a prognostic biomarker in patients with advanced NSCLC treated with ICIs.

## 1. Introduction

Non-small cell lung cancer (NSCLC) is the most frequent thoracic cancer in Europe and a leading cause of death both in Europe and worldwide [1]. Its incidence reached 2 million new cases in 2019. This cancer, especially in metastatic conditions, carries an extremely poor prognosis, with an overall 5-year survival rate of 13%. Non-squamous NSCLC (NS-NSCLC) is the most common histologic subtype [2].

With the rapid clinical development of monoclonal antibodies targeting the programmed death 1 pathway (PD-1/PD-L1,) also called immune checkpoint inhibitors (ICIs), the standard of care has rapidly evolved. Patients with metastatic NSCLC have been treated with chemotherapy followed by anti PD-1 mAb in recent years and ICI has become the standard first-line treatment of advanced NS-NSCLC in monotherapy for patients with a high expression of PD-L1 [3,4,5], and is associated with chemotherapy [6] for all comers with NSCLC.

Prediction of the response to ICI remains an unmet need. Although a large amount of studies have proposed many complex biomarkers, PD-L1 remains the only approved biomarker. Few studies have tested clinical biomarker that are easy to implement in clinical routines.

Myeloid-derived suppressor cells (MDSCs) can support tumor progression and have been shown to accumulate in the blood and peripheral lymphoid organs, such as the spleen, in animal models of cancer, leading to splenomegaly. We and others described an accumulation of circulating MDSCs in patients with metastatic colorectal cancer (mCRC) [7,8] and in those with pancreatic cancer as compared to healthy donors [9,10,11]. A high level of circulating MDSC at baseline is significantly associated with poor progression-free survival (PFS) and poor overall survival (OS) in mCRC and pancreatic cancer. High level of MDSC is also associated with resistance to ICI in preclinical models [12,13]. In a mouse model, splenomegaly was a surrogate marker of MDSC level and was associated with a poor prognosis [14]. We previously reported that baseline splenomegaly is a predictive marker of a poor response to FOLFIRINOX in advanced pancreatic carcinoma [15] and in metastatic colorectal cancer treated with FOLFIRI in the first line [16]. In addition, we demonstrated a correlation between splenomegaly and blood MDSC number.

In this retrospective study, we aimed to determine the prognostic role of splenic volume in metastatic NSCLC patients treated with ICIs. Secondly, we aimed to determine whether splenic volume is correlated with other classical prognostic markers.

## 2. Materials and Methods

### 2.1. Study Population

Four hundred five patients with metastatic lung cancer receiving a treatment with anti-programmed death 1 (PD-1/PD-L1) checkpoint inhibitors in the Georges François Leclerc Cancer Center between 2014 and 2020 were selected. Patients receiving immunotherapy treatment in adjuvant conditions were excluded.

Only patients whose splenic volume could be determined at baseline and after two months of immunotherapy treatment were retained for the analysis. A total of 276 patients were finally included.

Demographics; cancer history; and pathological, clinical, biological, and radiological data (tumor response according to Response Evaluation Criteria in Solid Tumors (RECIST) v1.1 criteria [17], baseline spleen volume, and spleen volume during immunotherapy), as well as treatment outcomes, were all retrospectively collected from the medical records.

At the time of their first visit to the Department of Medical Oncology, all patients with cancer signed a general informed consent document. This consent allowed us to use their clinical and biological data in the cohort study. Only patients from whom informed consent was obtained and recorded in the medical chart were included in this retrospective study. No additional specific informed consent for this study was necessary.

### 2.2. PD-L1 Expression Analysis

PD-L1 protein expression in tumor cells was assessed using immunohistochemistry with a ready-to-use PDL1 commercial kit or with QR1, 22C3 antibodies, on the pathology platform of Dijon. PD-L1 positivity was defined as >1% of cells in tumors.

### 2.3. Spleen Volume Estimation

Spleen volume was measured by means of CT scanning, as previously described [18]. The maximal width (W) of the spleen, determined as the largest diameter on any transverse section; the thickness at the hilum (Th), determined as the distance between the inner and outer borders of the spleen on a plane perpendicular to the splenic width and through the hilum; and the length (L) were obtained from abdominal CT examinations. Spleen volume was calculated using the formula: Spleen volume = 30 + 0.58 (W × L × Th). Spleen measurements and volume calculations were performed by the same investigator, who was blinded to clinical outcomes.

### 2.4. LIPI Score Calculation

Blood cell counts and LDH levels at baseline before immunotherapy treatment (within 30 days before the first treatment) were extracted from electronic medical records. Demographic, clinical, pathological, and molecular data were also collected.

The lung immune prognostic index (LIPI) was developed on the basis of dNLR (leukocytes/(leukocytes—neutrophils) >3) or LDH > 230 U/L (considered as the upper limit of the normal in our center) [19]. We considered two distinct groups: negative if none of these two conditions are observed, and positive if one or the two conditions were presented.

### 2.5. RNAseq Data

Total RNA was extracted from formalin-fixed paraffin-embedded (FFPE) tumor slices using the Maxwell 16 LEV RNA FFPE Purification kit (Promega, Madison, WI, USA), following the manufacturer’s instructions. Libraries were prepared from 12 µL of total RNA with the TruSeq Stranded Total RNA using Ribo-Zero (Illumina, San Diego, CA, USA), following the manufacturer’s instructions. Once qualified, paired-end libraries were sequenced using a 2 × 75-bp output on a NextSeq 500 device (Illumina).

The abundance of transcripts from RNA-seq data were quantified through the Kallisto program [20]. This program is based on a pseudo-alignment process for rapidly determining the compatibility of reads with targets, without the need for alignment. The Kallisto transcript index used as reference was built from merged human cDNA and ncDNA files from GRCh37 assembly ENSEMBL. Gene-level count matrices were created with the DESeq2 library. Low-count genes were pre-filtered by removing genes with too few reads [21].

### 2.6. Statistical Analysis

The evaluation of the tumor response was performed after four ICI injections. Progression-free-survival (PFS) was calculated from the date of first immunotherapy administration until disease progression or death to any cause, and was evaluated at 6 months. Overall survival (OS) was calculated from the date of first immunotherapy administration until death to any cause and was censored at 12 months.

Disease characteristics were examined using the Chi-squared test or Fisher’s exact test for qualitative variables and the Mann–Whitney test for continuous variables, as appropriate. Splenic volumes were analyzed as continuous or dichotomous variables and their prognostic role in relation to OS and PFS were determined using univariate and multivariate Cox models. Dichotomization of the variables was performed following the methodology of Lausen et al. [22]. Variables with *p*-values < 0.10 as determined by univariate analysis were entered into the multivariate model. Survival probabilities were estimated using the Kaplan–Meier method and survival curves were compared using the log-rank test. All *p*-values inferior to 0.05 were considered statistically significant.

Genes that were differentially expressed, given a decrease or increase in splenic volume, were compared and selected using the DESeq2 R package (Love et al.) [21]. Gene set enrichment analysis (GSEA) [23] was performed on the resulting differential genes using the C7 and Hallmark gene sets from the Broad Institute and the R package “clusterProfiler” [24].

Statistical analyses were performed using R software (http://www.R-project.org/, version 4.0.3, 1 October 2020) and graphs were drawn using GraphPad Prism version 7.0.3 (GraphPad, San Diego, CA, USA).

## 3. Results

### 3.1. Patients’ Clinical Characteristics

Out of the 405 patients treated with immune checkpoint inhibitors (in the first line or thereafter) for NSCLC between January 2014 and June 2020 in our center, 109 patients were excluded because their CT scans were not amenable to measurement of the spleen because of missing data, technical problems, or splenectomy (Figure 1). In total, the analysis was performed only on 276 patients with complete clinical and radiological information.

The mean age was 65 years. One hundred seventy (62%) patients had non-squamous NSCLC, 82 (30%) had squamous NSCLC, and 24 (8%) had other histological subtypes. Fifty-nine patients (20%) had KRAS mutations. PD-L1 status was positive in 146 patients (53%), negative in 66 (24%), and not available in 64 (23%) patients. Only four patients in our cohort had a concomitant treatment with granulocyte colony-stimulating factor (GCSF), and three patients had a history of chronic infection (two with B hepatitis and one with C hepatitis, without circulating virus). None had active infection (active B or C hepatitis, tuberculosis, candidemia).

The main characteristics of the population are reported in Table 1.

Splenic volume was analyzed before and at the moment of the first evaluation. The median splenic volume change during follow up was 4.39% (ranging from −69% to +81%) (Figure 2A). The treatment with ICI resulted in an increase in spleen size in 188 (63.5%) patients and a decrease in 108 (36.5%) patients.

In the whole cohort, median dNLR was 2.3 (interquartile range (IQR) = 1.6) and was greater than three in 102 patients. Median LDH was 217 (IQR = 86.8) and was greater than the upper limit of normal (ULN = 230) in 73 patients. One hundred thirty-two (47.8%) patients presented a LIPI score > 0. We observed a significant positive correlation between LDH and splenic volume at baseline (*p* = 0.02), volume during treatment (*p* < 1.10–3), and variations during treatment (*p* = 0.003) in splenic volume (Figure 2B). A significant correlation was also observed between dNLR and variations in splenic volume during treatment (*p* = 0.009), but not with baseline volume or splenic volume during treatment.

### 3.2. Associations between Splenic Volume and Progression-Free Survival (PFS) or Overall Survival (OS)

Median PFS was 3.7 (3.1; 4.9) and median OS was 16.1 (13.2; 18.7). We analyzed baseline splenic volume and volume during treatment as binary variables. Variables were dichotomized using the methodology of Lausen et al. [22].

The baseline splenic volume threshold retained for PFS and OS was 194 mL. In our cohort, the baseline splenic volume was ≤194 mL in 119 patients (43%), and >194 mL in 157 patients (57%). Using this threshold in univariate Cox analysis, baseline splenic volume was identified as a prognostic factor for OS (HR = 2 (1.3; 3.1); *p* = 0.001) (Figure 3A).

Based on multivariate Cox analysis, adjusting for cerebral, liver, pleuro-peritoneal and bone metastasis, PDL-1, number of treatment lines and LIPI score, the baseline splenic volume threshold of 194 mL remained significantly associated with OS (HR = 2.6 (1.44; 4.86); *p* = 0.002) (Table 2).

Using this same threshold, baseline splenic volume was not significantly associated with PFS by univariate Cox analyses (HR = 1.3 (0.9; 1.8); *p* = 0.1) (Figure 3D) or by multivariate Cox analyses (1.25 (0.86; 1.82); *p* = 0.241) (Table 3).

Considering the splenic volume during treatment, for the evaluation of PFS, we retained the threshold of 209 mL (119 patients with volume ≤ 209 mL and 157 > 209 mL). Patients with splenic volume > 209 mL during treatment had significantly poorer PFS than patients with a volume ≤ 209 mL, according to univariate and multivariate analyses (respectively, 1.4 (1.0; 1.9); *p* = 0.02 and HR = 1.4 (1.0–2.1); *p* = 0.05) (Figure 3B). The same findings were observed in univariate concerning this volume and a threshold of 221 mL for OS with HR = 2.2 (1.5; 3.3); *p* < 1.10^−3^. Splenic volume during treatment was no longer significant according to the multivariate Cox model 1.2 (0.7; 2.1); *p* = 0.4 (Figure 3E).

The change in splenic volume was also associated with PFS and OS. For PFS, HR = 1.8 (1.3; 2.6); *p* = 0.001 in the univariate Cox model, and when adjusted for clinical variables, HR = 1.3 (0.8–2.1); *p* = 0.2. For OS, HR = 2.3 (1.5; 3.8); *p* < 1.10^−3^ in the univariate Cox model and when adjusted for clinical variables, HR = 2.1 (1.1–3.8); *p* = 0.01. (Figure 3C,F).

### 3.3. Transcriptomic and Histological Features Related to Splenic Volume

To link this clinical observation with a transcriptomic molecular pattern, we analyzed RNAseq data in a subset of 50 patients included in this cohort. At the immunological level, GSEA analysis showed the significant enrichment of gene sets associated with the inflammatory response for patients with an increase in splenic volume (inflammatory response (*p* < 1.10^−3^), phagocytophilum stim neutrophil (*p* < 1.10^−3^)). Patients with an increasing splenic volume also significantly over-expressed genes associated with the TNF-alpha signaling pathway (*p* < 1.10^−3^) and under-expressed genes associated with the monocyte vs neutrophil pathway (*p* = 0.001) (Figure 4).

## 4. Discussion

This study demonstrates that splenic volume could be a predictive marker of PFS and OS in patients treated with ICIs in metastatic NSCLC. As splenic volume is known to be a surrogate marker of MDSC levels, this study raises the hypothesis that MDSC could modulate the efficacy of immunotherapy in metastatic NSCLC. Our study is the first to observe that splenic volume is associated with PFS in metastatic NSCLC. It confirms our previous findings in patients treated with a FOLFIRINOX regimen or FOLFIRI plus bevacizumab, respectively, in advanced pancreatic carcinoma and in metastatic colorectal cancer [15,16].

Baseline splenic volume is associated with OS but not PFS and thus can be considered a prognostic marker. The splenic volume during treatment and the percentage change in splenic volume between the start of treatment and the first evaluation via CT scanning, used as binary variables, were associated with PFS and OS and may provide a new predictive marker of ICI efficacy for metastatic NSCLC, insofar as we show that a splenic volume > 209 mL at first the CT evaluation is associated with poor PFS. Nevertheless, additional studies are required to validate this cut-off, and also to determine whether it is a new prognostic marker or predictor of the efficacy of immunotherapy.

The neutrophil/lymphocyte ratio present at the start of immunotherapy, which could be a surrogate marker of systemic inflammation, was also demonstrated to be associated with survival [25,26]. A lung immune prognostic index (LIPI) based on a dNLR ratio > 3 and a LDH above the ULN was developed by Mezquita et al. [27], characterizing three prognostic groups (good, zero factors; intermediate, one factor; poor, two factors). In a large cohort of NSCLC patients treated in second line or after with ICI, they demonstrated the predictive role of this marker and observed that systemic inflammatory status was closely correlated with a worse prognosis in lung cancer. Similarly, additional reports have shown the prognostic and predictive role of NLR or LDH levels in either lung cancer or melanoma treated with ICI [19,28,29]. We previously showed (manuscript submitted) that LIPI score was associated with poor dendritic cell tumor infiltration and immunosuppression. Our data linking LIPI and splenic volume suggest that LIPI is a surrogate marker of MDSC levels.

Numerous studies have shown a link between chronic inflammation and different cancers. Such inflammation could promote cancer growth and could negatively affect the immune system by inducing immune subversion. Such observations could be related to pathological myelopoiesis, which induces the accumulation of myeloid-derived suppressor cells (MDSCs) [30,31]. These cells are a population of immature myeloid cells close to neutrophils, and are present in the circulation and tumors of patients with cancer, and they play a role in the inhibition of the immune response against cancer. MDSCs can support tumor progression by promoting tumor cell survival, angiogenesis, the invasion of healthy tissue by tumor cells, and metastasis [32,33,34]. In animal models, the accumulation of MDSCs is associated with splenomegaly [35]. A similar correlation was also observed in hepatocellular carcinoma [36].

We previously reported that high baseline levels of MDSCs are associated with shorter PFS in metastatic colorectal cancer patients on the FOLFOX regimen [8] and that MDSC level is correlated with splenic volume. These data support the notion that splenomegaly could be a surrogate marker of MDSC accumulation. Although MDSCs are powerful immunosuppressive cells, they exert their deleterious effect via their capacity to blunt the T-dependent antitumor immune response [31]. These data suggest that splenic volume is a surrogate marker of MDSC-dependent immunosuppression, and therefore it may explain the negative effect of volume change during immunotherapy treatment, and the negative effect of high splenic volume at baseline or during treatment on ICI efficacy.

The limitations of this study include the retrospective nature of the data and the presence of missing clinical and pathological data, albeit only for a low proportion of patients. The measurement of splenic volume is rapid and easy to perform but could lack reproducibility. Another possible bias relates to the pooling of patients treated with different ICI mAbs at different lines of treatment. Moreover, it would have been interesting to correlate splenic volume at different time points and macroscopic lesions or metabolic parameters on CT and TEP scans, respectively, but this was not carried out in this first exploratory study. External validation in other cohorts is needed to confirm the relevance of our results.

## 5. Conclusions

To conclude, our data reveal that the evolution of splenic volume during immunotherapy regimens could be used to stratify and predict PFS and OS in patients treated for NSCLC. Splenic volume seems to be linked to the LIPI score and MDSC accumulation, which are known to be associated with resistance to immunotherapy. These results warrant confirmation in other future prospective clinical trials with immunotherapy.

## Figures and Tables

**Figure 1 cancers-13-03020-f001:**
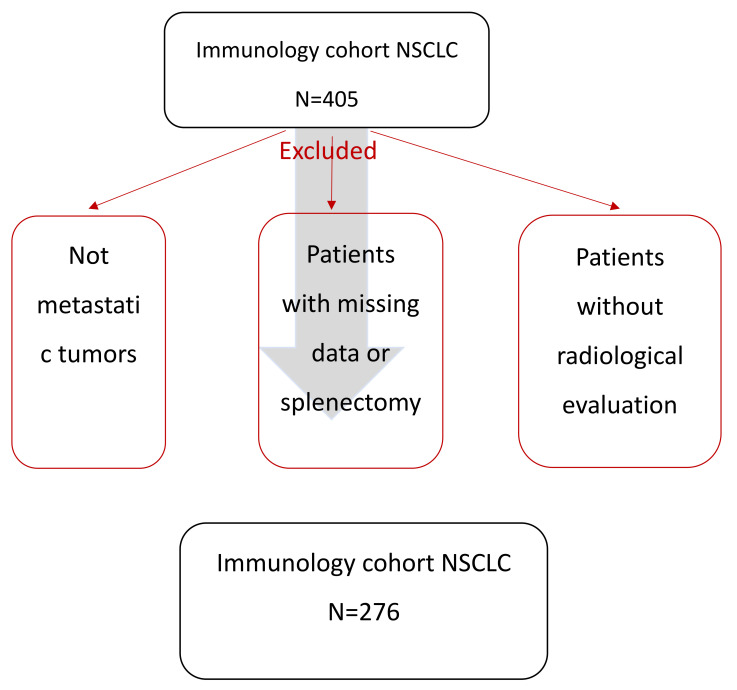
Study flowchart.

**Figure 2 cancers-13-03020-f002:**
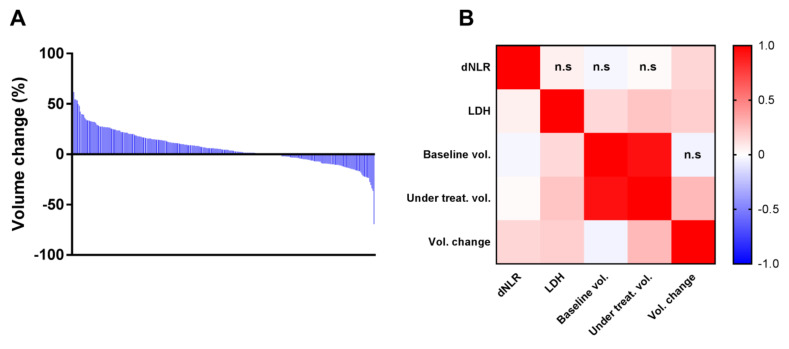
(**A**) Barplot representing decreasing volume changes for all of the patients. (**B**) Heatmap showing Pearson correlation matrix coefficients between splenic volumes, dNLR, and LDH measures n.s: non-significant correlation.

**Figure 3 cancers-13-03020-f003:**
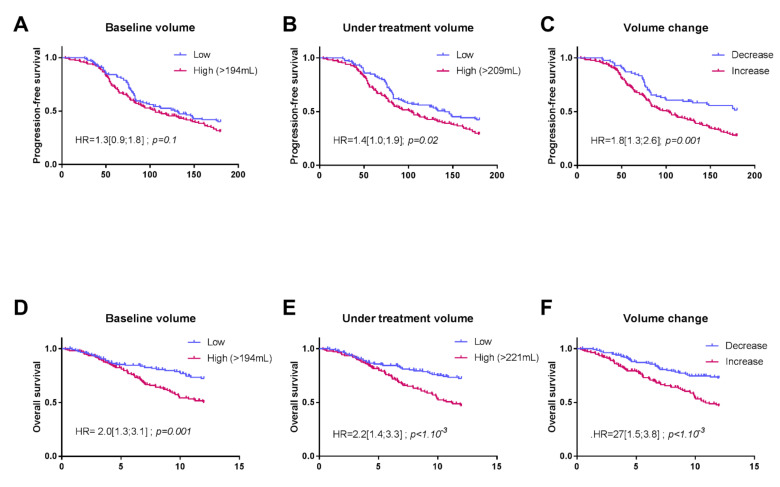
Kaplan–Meier curves for progression-free survival (PFS) (**A**–**C**) and overall survival (OS) (**D**–**F**), with patients stratified according to their splenic volume in the following order: baseline volume, volume during treatment, and volume change.

**Figure 4 cancers-13-03020-f004:**
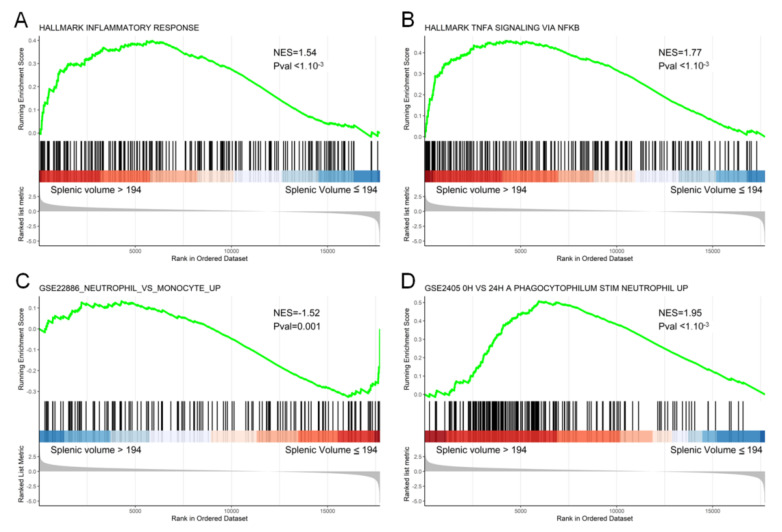
Gene set enrichment analysis (GSEA) enrichment plot for the proliferation of the inflammatory response (**A**), TNFA (**B**), neutrophil vs. monocyte (**C**) and phagocytophilum stim neutrophil (**D**) pathways at 24 h. The y-axis represents enrichment score (ES) and the x-axis shows genes (vertical black lines) represented in different pathways. The colored band at the bottom represents the degree of correlation of the expression of these genes (red for a high gene expression and blue for a low gene expression).

**Table 1 cancers-13-03020-t001:** Summary of clinical characteristics.

Variables	All Patients (N = 276)
Sex, n (%)	
Male	193 (69.9)
Female	83 (30.1)
Age at diagnosis, years, median (IQR)	65.4 (13.3)
n (%)	
≤60	86 (31.2)
>60	190 (68.8)
Smoking status, n (%)	
Never smoker	17 (6.2)
Current or former smoker	235 (85.1)
NA	24 (8.9)
Histological type, n (%)	
Non-squamous Squamous	170 (61.6) 82 (29.7)
Other	24 (8.7)
WHO performance status, n (%)	
0	80 (29)
>0	184 (66.7)
NA	12 (4.3)
Cerebral metastasis, n (%)	70 (25.4)
Liver metastasis, n (%)	66 (23.9)
Bone metastasis, n (%)	116 (42)
Lymph node metastasis, n (%)	206(74.6)
Pleuro-peritoneal metastasis, n (%)	169 (61.2)
Line of ICI, n (%)	
1	94 (34)
>1	182 (66)
Type of ICI, n (%)	
Anti PD-L1	34 (12.3)
Anti PD-1	218 (79)
Anti CTLA4	1 (0.4)
Anti PD-L1 + anti CTLA4	4 (1.4)
Anti PD-1 + anti CTLA4	9 (3.3)
Other association	10 (3.6)
PD-L1 status (cutoff at 1%), n (%)	
Negative tumors	66 (23.9)
Positive tumors—Strong expressors	146 (52.9) –75
NA	64 (23.2)
LDH, median (IQR)	217 (86.8)
dNLR, median (IQR)	2.3 (1.6)
LIPI score, n (%)	
0	98 (35.5)
>0	132 (47.8)
NA	46 (16.7)
OS (months) (95% IC)	16.1 (13.2; 18.7)
PFS (months) (95% IC)	3.7 (3.1; 4.9)

ICI: immune checkpoint inhibitors; IQR: interquartile range; WHO: World Health Organization; LDH: lactate dehydrogenase; dNLR: derived neutrophils/(leukocytes minus neutrophils) ratio; LIPI: lung immune prognostic index; NA: not available.

**Table 2 cancers-13-03020-t002:** Univariate and multivariate Cox models for overall survival (OS). Three multivariates were estimated, each involving either baseline volume, volume during treatment, or volume change.

Variables	Univariate	Multivariate		
		Baseline Volume	Volume during Treatment	Volume Change
Sex				
Female	1	1	1	1
Male	1.5 (1.0; 2.4) *p* = 0.06	0.8 (0.4; 1.5); *p* = 0.6	1.0 (0.5; 1.8); *p* = 0.9	0.9 (0.5; 1.7); *p* = 0.8
Age at diagnosis, years, median (IQR)		--	--	--
≤60	1			
>60	0.9 (0.6; 1.4); *p* = 0.7			
Smoking status		--	--	--
Never smoker	1			
Current or former smoker	1.8 (0.7; 4.9); *p* = 0.2			
Histological type				
Adenocarcinoma	1	1	1	1
Epidermoid	1.4 (0.9; 2.1); *p* = 0.1	2.1 (1.1.1); *p* = 0.03	2.1 (1.1; 4.0); *p* = 0.02	1.6 (0.8; 3.1); *p* = 0.2
Other	1.4 (0.4; 0.7); *p* = 0.4	0.7 (0.2; 3.4); *p* = 0.7	0.73 (0.1; 3.3); *p* = 0.7	0.7 (0.1; 2.9); *p* = 0.6
WHO performance status				
0	1	1	1	1
>0	2.8 (1.7; 4.8); *p* < 1.10^−3^	1.8 (0.9; 3.5); *p* = 0.07	1.8 (0.9; 3.4); *p* = 0.09	1.7 (0.9; 3.4); *p* = 0.1
Cerebral metastasis	1.4 (0.9; 2.2); *p* = 0.1	1.3 (0.7; 2.6); *p* = 0.4	1.3 (0.66; 2.7); *p* = 0.42	1.2 (0.6; 2.3); *p* = 0.6
Liver metastasis	2.6 (1.7; 3.9); *p* < 1.10^−3^	2.9 (1.7; 5.1); *p* < 0.001	2.5 (1.5; 4.4); *p* = 0.001	2.2 (1.3; 3.9); *p* = 0.004
Bone metastasis	1.7 (1.1; 2.5); *p* = 0.01	2.1 (1.2; 3.7); *p* = 0.01	1.83 (1.0; 3.3); *p* = 0.05	1.9 (1.0; 3.6); *p* = 0.04
Lymph node metastasis	1.3 (0.8; 2.1); *p* = 0.3	--	--	--
Pleuro-peritoneal metastasis	1.3 (0.9; 2.0); *p* = 0.2	1.2 (0.7; 2.2); *p* = 0.5	1.3 (0.7; 2.3); *p* = 0.4	1.4 (0.8; 2.4); *p* = 0.3
Line of ICI				
1	1			
>1	0.1 (0.04; 0.4); *p* < 1.10^−3^	0.2 (0.04; 0.7); *p* = 0.01	0.2 (0.04; 0.7); *p* = 0.01	0.2(0.04; 0.7); *p* = 0.01
PD-L1 status (cut-off at 1%)				
Negative tumors	1	1	1	
Positive tumors	0.6 (0.4; 1.0); *p* = 0.09	0.7 (0.4; 1.13); *p* = 0.1	0.7 (0.37; 1.20); *p* = 0.2	0.5 (0.3; 0.9); *p* = 0.03
LDH	1 (1; 1); *p* = 0.003	--	--	--
dNLR	1.2 (1.1; 1.3); *p* < 1.10^−3^	--	--	--
LIPI score				
0	1	1	1	1
>0	2.0 (1.3; 3.1); *p* = 0.003	1.4 (0.8; 2.5); *p* = 0.2	1.4 (0.8; 2.4); *p* = 0.3	1.3 (0.8; 2.3); *p* = 0.3
Baseline volume				
≤194 mL >194 mL	1	1	--	--
	2.0 (1.3; 3.1); *p* = 0.001	2.6 (1.4; 4.9); *p* = 0.002		
Volume during treatment				
≤221 mL >221 mL	1	--	1	--
	2.2 (1.45; 3.3); *p* < 1.10^−3^		1.2 (0.7; 2.1); *p* = 0.4	
Volume change				
≤4 >4	1	--	--	1
				2.10 (1.1–3.8); *p* = 0.01
	2.27 (1.5; 3.8); *p* < 1.10^−3^			

ICI: immune checkpoint inhibitors; IQR: interquartile range; WHO: World Health Organization; LDH: lactate dehydrogenase; dNLR: derived neutrophils/(leukocytes minus neutrophils) ratio; LIPI: lung immune prognostic index; NA: not available.

**Table 3 cancers-13-03020-t003:** Univariate and multivariate Cox models for progression-free survival (PFS). Three multivariates where estimated, each involving either baseline volume, volume during treatment, or volume change.

Variables	Univariate	Multivariate		
		Baseline Volume	Volume during Treatment	Volume Change
Sex				
Female	1	1	1	1
Male	1.5 (1.1; 2.1); *p* = 0.03	1.0 (0.7; 1.6); *p* = 0.9	1.0 (0.7; 1.6); *p* = 0.9	1.1 (0.7; 1.6); *p* = 0.7
Age at diagnosis, years, median (IQR)		--	--	--
≤60	1			
>60	0.9 (0.7; 1.2); *p* = 0.5			
Smoking status		--	--	--
Never smoker	1			
Current or former smoker	0.7 (0.4; 1.3); *p* = 0.3			
Histological type		--	--	--
Adenocarcinoma Epidermoïd	1 1.2 (0.9; 1.6); *p* = 0.4			
Other	0.8 (0.5; 1.5); *p* = 0.5			
WHO performance status		--	--	--
0	1			
>0	1.2 (0.84; 1.6); *p* = 0.3			
Cerebral metastasis	1.2 (0.9; 1.7); *p* = 0.3	--	--	--
Liver metastasis,	2.1 (17; 2.9); *p* < 1.10^−3^	1.9 (1.2; 2.80); *p* = 0.003	1.8 (1.2; 2.8); *p* = 0.003	1.8 (1.2; 2.7); *p* = 0.005
Bone metastasis	1.5 (1.1; 2.0); *p* = 0.01	1.5 (1.1; 2.2); *p* = 0.02	1.53 (1.1; 2.2); *p* = 0.02	1.51 (1.0; 2.2); *p* = 0.03
Lymph node metastasis	1.1 (0.8; 1.6); *p* = 0.7	--	--	--
Pleuro-peritoneal metastasis	1.2 (0.9; 1.7); *p* = 0.04	1.38 (0.9; 2.0); *p* = 0.1	1.35 (0.9; 2.0); *p* = 0.1	1.36 (0.9; 2.0); *p* = 0.1
Line of ICI				
1	1	1	1	1
>1	0.06 (0.02; 0.2); *p* < 1.10^−3^	0 0.06 (0.02; 0.2); *p* < 0.001	0.06 (0.02; 0.2); *p* < 0.001	0.06 (0.02; 0.20); *p* < 0.001
PD-L1 status (cut-off at 1%)				
Negative tumors	1	1	1	1
Positive tumors	0.7 (0.5; 0.9); *p* = 0.03	0.6 (0.4; 0.9); *p* = 0.01	0.6 (0.4; 0.9); *p* = 0.01	0.6 (0.4; 0.9); *p* = 0.01
LDH	1.0 (1.0; 1.0); *p* < 1.10^−3^	--	--	--
dNLR	1.1 (1.0; 1.2); *p* = 0.4	--	--	--
LIPI score		--		
0	1		--	--
>0	1.0 (0.7; 1.4); *p* = 0.9			
Baseline volume				
≤194 mL >194 mL	1		--	--
	1.3 (0.9; 1.8); *p* = 0.1	1.2 (0.9; 1.8); *p* = 0.2		
Volume during treatment		--		--
≤209 mL >209 mL	1			
			1.45 (1.0; 2.1); *p* = 0.05	
	1.4 (1.0; 1.9); *p* = 0.02			
Volume change		--	--	
≤0 >0	1			
	1.8 (1.3; 2.6); *p* = 0.001			1.3 (0.8; 2.1); *p* = 0.2

ICI: Immune checkpoint inhibitors; IQR: interquartile range; WHO: World Health Organization); LDH: lactate dehydrogenase; dNLR: derived neutrophils/ (leukocytes minus neutrophils) ratio; LIPI: lung immune prognostic index; NA: not available.

## Data Availability

The datasets analyzed during the current study are available from the corresponding author upon reasonable request.

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
