# Peer review of "Splenic Volume as a Surrogate Marker of Immune Checkpoint Inhibitor Efficacy in Metastatic Non Small Cell Lung Cancer"

_cancers, 2021, doi:10.3390/cancers13123020_

Round 1

Reviewer 1 Report

Dear Authors,

I have read a manuscript with interest.

In the manuscript considering issue can have a relevance for clinical practice.

Important shortcomings that could be approved according to the Review process>

It is important that only patients treated with mono-immunotherapy without chemotherapy and radiation treatment combinations will be included in the analysis. Both, previuos chemo- and radiotherapy can significantly influence splenic volume. Please, verify and point out in the text.

It is also important to exclude patients with history of GCSF (granulocyte colony-stimulating factor) analoga treatment and with onset of sepsis events or chronic infections (viral, bacterial, mycoses, pleuralempyema ets) from analysis. Please, verify and point out in the text.

It will be important to correlate splenic volumes at differnt time Points with macroscopic Tumor burden. At least, maximal number of lesions, sume of maximal diameter of lesions and  if appropriate metabolic parameters (MTV, SUVmax, TLG) could be analyzed regarding to initial and changes of splenic volume under treatment.

A clear hypothesis > why dynamic changes of splenic volume can impact an efficacy of immune check-point Inhibition > must be formulated in the Discussion and Conclusion chapters.

Reviewer 2 Report

The present manuscript reports on retrospective study evaluating the potential role of splenic volume as marker of immune checkpoint inhibitors (ICIs) efficacy. This is a very novel topic and could add further evidence on the role of clinical/laboratory findings on ICIs activity in NSCLC.

The most significant finding reported here is that splenic volume might represent a surrogate marker of MDSC-dependent immunosuppression, explaining the negative effect of high splenic volume on ICIs efficacy.

Some comments:

  • How was defined splenic volume change? Did the authors used a cut-off value? Do change during treatment refer to the first radiological evaluation or even to subsequent CT scans and/or every CT restaging? The same considerations can be applied for dNLR, LDH and LIPI changes during treatment. Please clarify
  • Most of the patients were treated with single agent PD-1/PD-L1 inhibitors. Did the authors conduct a separate analysis restricted to these patients only, as the patients treated with combinations is highly heterogenous?
  • Table 1 should also contain the data of PD-L1 strong expressors.
  • It is surprising that 23 patients had no reported/unknown treatment line, according to table 1. Please clarify
  • Table 1 misses data of TNM stage
  • Figure 1 was likely uploaded incorrectly (the red lines should not be present, I suppose).

Round 2

Reviewer 2 Report

The authors have positively replied to my comments/suggestions. I have no further comments